# Lessons from Shawshank: Outlaws, Lawmen and the Spectacle of Punishment

**Benjamin Boyce**

Communication Department, University of Colorado Denver, Denver, CO 80204, USA;
benjamin.boyce@ucdenver.edu

**Abstract:** For more than a century, cinema has offered a rich source of images and narratives about crime and punishment. Unfortunately, the restricted nature of correctional environments and the social stigma surrounding incarceration leave most viewers reliant on media representations for the majority of their knowledge about correctional spaces. In most media representations of crime and punishment, outlaws and lawmen are reduced to stereotypical archetypes, and incarcerated characters are some of the evilest villains one will ever encounter. Moreover, the prison environment is painted as a playground for bad behavior, as penance for redeemable outlaws, or as an outright paradox that claims to reduce criminality despite appearing to increase it. Our uncritical acceptance of such characterizations goes hand in hand with our cultural addiction to mass incarceration. Limitless stories about uncontainable monsters perpetrating awful crimes inside cushy taxpayer funded facilities endorse a worldview where a permanently expanding and harshening prison system is vital to the safety of a functioning society. In short, our reliance on the spectacle of punishment has left us woefully and willfully misinformed about prison and those who wind up there.

**Keywords:** Shawshank; Litchfield; *Oz*; *Les Misérables*; incarceration; spectacle; television; stereotypes; archetypes; prison cinema





## 1. Introduction

In the fall of 2004, I (the author) was shackled to a group of a dozen other men, loaded onto a transport van, and shipped to what was the largest walled prison on earth when it began housing incarcerated people in 1937 (Bright 1996). That long trip to Jackson, Michigan, on the opposite side of the state from where I grew up, was the final chapter in my preparation for prison—my dress rehearsal for the penitentiary. I knew what was coming. I had seen sexual assaults and requisite violence depicted in *Blood In Blood Out* (Hackford 1993). The sound of the entire cellblock in *The Shawshank Redemption* chanting "Fresh fish!" until one man breaks down crying for his mother rang in my ears (Darabont 1994). The murder of a suspect just before he is released in *Short Eyes* replayed in my mind's eye (Young 1977). I believed I was about to enter a world of corrupt officials, heartless administrators, and cruel monsters all intent on victimizing me. Those were the only images of prison I had ever seen, so I had prepared for those images. I did hundreds of pushups every day to bulk up my skinny frame. My weekly order of snacks grew as I tried to pack on the pounds. The soft expressions and light tone I normally used to communicate with those around me were repressed in favor of a hard glare and a deep voice. I transformed myself into what I knew prison would require me to be. Then I got there and realized everything I thought I knew about prison was wrong, or at least misunderstood. The spectacle of punishment, from television and films to commercials and social media, had convinced me to prepare for a specter that never appeared. But how did that happen?

In the following pages, I track the evolution of cinematic spectacles of punishment, focusing on what Michelle Brown has called, "a shadow world of moral judgement and penal logics [that] exists beyond prison walls as a constant and perpetually growing cultural

resource for people to make sense of punishment" (Brown 2009, p. 5). In the United States, those who have never been to prison are afraid of those who have, a fear learned and reinforced through the stories we consume (Yousman 2013). I end with a call for cultural and cinematic reorientation. The recipe for cinematic prisons has long required certain key ingredients which titillate audiences while leaving us creeped out about the bad guys we see inside. Until the representation is updated, those who have never been to prison will continue to look at those who have as dangerous outsiders with warped morals. Until we change the way we think about those in prison, we will continue to demand the same problematic spectacles as always. Our cultural attitudes regarding crime and punishment must change alongside our expectations for media productions.

## 2. As Seen on TV

Cinema has been teaching viewers about the world for more than a century, and cultural critics have been there from the start, pointing out the dangers of our addiction to the spectacle. Walter Benjamin diagnosed our unhealthy affection for the "phony spell of commodity" in 1935, before most homes contained a television (Benjamin 1935, p. 231). By 1947, Theodor Adorno and Max Horkheimer were criticizing our cultural habits of "aesthetic mass consumption," noting the ways that media spectacles designed to appeal to a mass audience often teach us things about the world that are not necessarily true (Adorno and Horkheimer 1947, p. 139). At the time, the world was still coming to terms with the atrocities of the Holocaust, and cultural critics were there to pinpoint the relationship between mediated messages and human motivation. In 1963, after attending Adolf Eichmann's trial for crimes against humanity and the Jewish people, Hannah Arendt described Nazi war criminals as having been influenced by mediated messages that caused them to act in ways that were entirely predictable. All of us, Arendt claimed, are capable of doing the same thing if we are exposed to the same sorts of mediated messages.

As black and white TVs turned to HD flatscreens, and the rabbit ears antenna was swapped out for cable, cultural critics kept pace. In 1984, Michel de Certeau unpacked the ways media teach us things about the world even when we know we are consuming fiction: "the spectator-observer *knows* that they are merely 'semblances,' the results of manipulations . . . *but all the same*, he assumes that these simulations have the status of the real" (p. 188). By 1994, theory had moved to example as bell hooks described how media depictions of poverty teach viewers, wrongly, that the poor are immoral and driven by greed, and that wealth typically comes from hard work (p. 197). Naturally, those who watch a lot of television and films come to believe that wealth springs from ethics, and that most poverty is the result of laziness or immorality. Our beliefs inform our behavior.

The twenty-first century brought new statistics to strengthen well-established theory. Travis Dixon found that television news programs constantly overrepresent the number of crimes committed by non-white people and the number of arrests made by white officers, causing viewers to unwittingly accept the false narrative that crime is much more common than it actually is, and that it is usually committed by nonwhite suspects (Dixon 2017). Bill Yousman verified that watching prison-based television shows full of mayhem and murder unwittingly causes us to become more supportive of punishment-oriented policies, up to and including the death penalty (Yousman 2013). Kathleen Donovan and Charles Klahm IV described why those who watch television crime dramas are more likely to believe the police are generally successful at reducing crime, that they only use force when necessary, and that misconduct or abuse at the hands of police seldom leads to false confessions (Donovan and Klahm 2015). Allison Page and Laurie Ouellette reported that even when cinematic prisons feature dialogue and plot designed to challenge the norm of mass incarceration, we often still come to endorse an expanding and harshening correctional system based on the behavior of those we see inside (Page and Ouellette 2019).

None of these effects take place consciously within the viewer's mind. If you ask television or movie audiences how much their opinions have changed based on their media diet, they usually claim to be uninfluenced by the stories they consume and immune to the

messages that affect the rest of us (Yousman 2013; Dixon 2017). We frequently fail to notice the bulk of what we learn from media.

Meanwhile, Hollywood continues to sell us constantly repackaged facsimiles, a situation that becomes obvious when one examines the wares on offer from the spectacle. Nine Spiderman films have been produced and sold to the public since 2002 (Miller et al. 2021). Television is full of "new" productions based on shows that were canceled decades ago: *The Wonder Years, Saved by the Bell, CSI: Las Vegas, Quantum Leap, True Blood, Frasier, A League of their Own, Charmed, Will & Grace, (Fresh Prince of) Bel-Air*, and *Gossip Girls*, along with plenty of others. Reboots have become an increasingly popular solution to consumer demand for content that appears new while still scratching familiar itches. We want the freshness of novelty alongside the comfort of the common. The never-ending growth of new streaming services has only increased the rate of production, and social media has both expanded alongside the spectacle and worked to strengthen its attraction.

Media productions reflect our cultural norms, but they are also a reflection *of* cultural norms. They show us the truth about ourselves and how we think, but they also shape our truths and our thoughts. As bell hooks explains:

> Much of the magic of cinema lies in the medium's power to give us something other than life as is ... Movies do not merely offer us the opportunity to reimagine the culture we most intimately know on the screen, they create culture. (Hooks 1996, p. 12)

Of course, sometimes producers get it wrong. As cinematic spectacles venture from the mundane into the bizarre, viewers are often confronted with information that cannot be challenged with real life experience. When faulty information is the only information available, we come to accept fiction as fact. The stories in our movies and TV shows become the stories in our imaginations, and we often lack the experience necessary to counterbalance the spectacle. As Michelle Brown has suggested, "much of the popular knowledge about punishment is constructed—in spaces far from the social realities and the social facts that define mass incarceration" (Brown 2009, p. 4).

Seven-Block was a symphony of echoes when I got back from chow that first night. The cellblock is one of many spread across the 52-acre facility located on the outskirts of Jackson. I climbed the steel staircase to level 3, then started counting as I walked past cells; 1, 2, 3 ... mine was the 15th cell on the tier, halfway down the walkway. The unit was five stories tall with 150 cells on each side, and it sounded like all 300 of us were always talking all the time. At cell 7, I stopped and asked an old-timer if he had anything for me to read. He smiled and handed the first half of an Encyclopedia Brown novel through the bars. Every book at Jackson was ripped in half so two people could read it at the same time. There still weren't nearly enough books to go around. Writing from Death Row, Mumia Abu-Jamal once said, "Life here oscillates between the banal and the bizarre" (Abu-Jamal 1995, p. 6). The banal is the norm. Warehoused humans with wasted potential are packed from corner to corner. The spectacle of punishment seldom captures the boredom that predominates the US correctional environment. Two hours later, I finished the half-book and started reading it over again.

### 2.1. Introducing the Cast: Les Misérables

The latest English (language) movie reboot of Victor Hugo's (1862) outlaw story *Les Misérables* hit theatres in 2012, becoming an immediate success (Hooper 2012). The blockbuster made $283 million during its first month in theaters, winning three Golden Globes and three Academy Awards in the process (Wioszczyna 2013). Many were unsurprised with the film's popularity. Since 1985, the musical upon which the film is based has been translated into at least 21 languages, performed in no less than 43 countries, and won upwards of 100 awards while being seen by more than 60 million people (Dargis 2012). As early as 1909, movie producers were taking advantage of the story's notoriety (Blackton 1909). And to this day, there is something about this narrative that continues to resonate with audiences across the world.

The success of any cinematic production depends, in part, on the producers' use of archetypes—recurring characters in literature, film or television who are identifiable by commonly held traits. Much like the latest Hollywood reboot, each new production appears original at first glance, but closer inspection reveals repackaged hand me downs disguised as unique creations. As Carl Jung explained, "The archetype is a tendency to form such representations as a motif—representations that can vary a great deal in detail without losing their basic pattern" (Jaffé 1964, p. 58). Archetypes do more than just help us make sense of the stories we consume; they help us make sense of the world.

Cinematic archetypes make us feel comfortable. When they appear, we know what to expect, so once they set up shop in a culture's media representations, they stick around for a long time. As Aniela Jaffe suggests, "The intertwined history of religion and art, reaching back to prehistoric times, is the record that our ancestors have left of the symbols that were meaningful and moving to them" (Jung 1964, p. 257). Contemporary archetypes are the shadows and echoes of stories which were valued and shared by those who came before us. That means we can use archetypes to identify genealogical patterns in media which reveal underlying trends in cultural norms; we can decode them to better understand ourselves and our place in the world. In *Les Misérables*, we meet two archetypal characters whose shadows continue to reappear in film and literature to this day: the outlaw who is seeking redemption, and the lawman who refuses to extend it. As they materialize in different settings and plots, the outlaw and the lawman operate within an archetypal prison space that both reveals and shapes cultural understanding of prison and those inside it.

## 2.2. The Archetypal Outlaw

The protagonist in *Les Misérables* is Jean Valjean, an outlaw who is granted parole near the story's beginning.[1] He was originally arrested for stealing a loaf of bread to feed his family, but despite his best efforts to move beyond his conviction, the rest of the world appears unwilling to give him a second chance. Marked as a criminal, he struggles to find employment, food and housing. As viewers, we empathize, that is, until a stranger offers him shelter and he responds by making off with the man's silver in the middle of the night. Those who had turned Valjean away based on his record seem, for a moment, to have been correct in their assumptions about his character, as he proves himself to be a stereotypical recidivist. As the story picks up steam, we are invited to wonder whether this outlaw is capable of redemption at all.

Valjean is arrested and dragged back to the scene of the crime, but in a twist, the homeowner claims the items were a gift. The stranger's kindness is a turning point in the story, causing Valjean the incorrigible criminal to begin the transformation into Valjean the redeemed outlaw. As viewers, we become engrossed in a plot driven by emotions we seldom experience so richly in our own lives: relief, fear, anticipation, terror, and most importantly, redemption. Throughout the following years, Valjean makes us proud: he rescues a young girl from her abusers, he becomes Mayor of a local town, and he even saves the life of a stranger who becomes pinned beneath a cart. Redemption becomes his sole mission in life, and by the story's conclusion, Valjean is hard to hate, while the lawman who relentlessly chases him has taken on many of the traits we originally disliked in Valjean.

*Les Misérables* stands as what Mike Nellis has called "the foundational redemptive text" (Nellis 2009, p. 131). According to Nellis, Valjean's redemption is established through specific events: he "experiences guilt and takes responsibility for his future, makes amends by becoming a productive citizen and prevents suffering in another generation," all while enduring "whatever moral duty requires of him" (Nellis 2009, p. 132). This basic archetypal character of the outlaw seeking redemption has remained a prominent figure in prison cinema for more than a century. The level of success which the outlaw experiences in their attempts at reformation varies from story to story, and sometimes the archetypal outlaw discards the idea of redemption altogether. However, the desire for redemption always materializes at some point in the narrative of the redeemed outlaw.

### 2.3. The Archetypal Lawman

*Les Misérables* also introduced Police Inspector Javert, an archetypal lawman who represents the counterpart of the outlaw seeking redemption. Javert chases the reformed Valjean wherever he flees regardless of the years that pass or the change of heart the outlaw appears to exhibit. As Nellis explains:

> Javert, embodies all that is hostile to the principle of giving offenders second chances . . . he represents both the long shadow of imprisonment, and the impossibility of ever becoming free of its influence or stigma, and the diffusion of penal authority into every crevice of public life. (Nellis 2009, p. 132)

True to character, Javert cannot reconcile the outlaw's redemption with his belief that outlaws are irredeemable, so when Valjean risks his freedom and safety to save the inspector, Javert responds by taking his own life. The archetypal lawman lacks the capacity to comprehend a redeemed outlaw. When faced with the reality of a world where outlaws are capable of redemption, the lawman in *Les Misérables* simply cannot go on living.

### 2.4. Repackaged and Resold

The two archetypes detailed above—the outlaw seeking redemption and the lawman who refuses to offer it—continue to make regular appearances in contemporary media. They are not hard to spot, despite the best efforts of producers to disguise them as new and original in each production. And they go all the way back to the original moving pictures, carving a trail through the genealogy of cinema that reflects cultural norms and attitudes along the way.

In the early years of cinema, outlaw stories were generally simple narratives of redemption without much spectacle. In 1920, Buster Keaton played an archetypal outlaw in the silent film *Convict 13*, where his character is arrested and imprisoned, but eventually released after he proves his innocence (Cline and Keaton 1920). Films in which the outlaw's innocence is eventually established after a period of doubt make it easy for audiences to offer redemption, and as such, they have always been a popular choice for producers. In 1932, Richard Walters was a redeemed outlaw in *The Last Mile*, sent to death row for a crime he did not commit and surrounded by archetypal lawman and other incarcerated characters who all refuse to accept his claims of innocence until he is eventually pardoned (Bischoff 1932). A few years later, in 1934, Hollywood turned out an early reboot with *The Count of Monte Cristo*, which pitted redeemed outlaw Edmond Dantès against a warden, a prosecutor, and a prison structure that all seem dead set on keeping him locked up and miserable for the rest of his life (Lee 1934). The 1939 film *Convict's Code* followed outlaw Dave Tyler to prison for a crime he did not commit; once there, the warden and his parole officer encourage him to stop professing his innocence all the way up to the day he is eventually acquitted (Hillyer 1939). In 1940 it was mobster Tommy Gordan in Sing Sing prison in *Castle on the Hudson*, a film that never awards the outlaw his redemption despite his innocence; he is eventually put to death (Litvak 1840).

By the 1950s, Hollywood was catching its stride and beginning to reflect evolving cultural attitudes regarding civil rights, social justice, and incarceration. The 1957 film *Jailhouse Rock* follows redeemed outlaw Vince Everett, played by Elvis Presley, into prison and through his efforts to become a musician after his release, an accomplishment which gains him what Mike Nellis calls "redemption through art" (Thorpe 1957; Nellis 2009, p. 140). As the 1960s Civil Rights Movement picked up steam, the possibility of redemption being an option for even the most culturally feared members of society became central to many spectacles of punishment. *The Bird Man of Alcatraz* hit theatres in 1962, loosely based on the life of redeemed outlaw Robert Stroud, who is sent to prison for murder but eventually finds redemption through personal growth and humanizing acts of assistance, including rescuing and raising birds and getting others in prison to do the same. (Frankenheimer 1962). In 1967, *Cool Hand Luke* introduced movie-goers to veteran Luke Jackson, who is sentenced to prison for drunkenly stealing parking meters and eventually killed by the obstinate lawman when he tries to escape (Rosenberg). Viewers were also treated to Robert

Aldrich's *The Dirty Dozen* in 1967, a story about a group of twelve redeemed outlaws, each sentenced either to life in prison, or to death, and each offered a deal whereby they can earn their freedom (and their redemption) if they accept a dangerous mission to kill a group of Nazis (Aldrich). The outlaws and lawmen in these productions reflected an optimistic cultural belief that those in prison were capable of redemption, but only if the system was set up to foster it. Despite the apparent updates, just like earlier cinematic representations of crime and punishment, these films continued to utilize the tried-and-true archetypal outlaw and lawman operating within an archetypal prison.

By the 1970s, cultural support for updating the criminal justice system to focus on rehabilitation was largely replaced by a cynical attitude that change was not possible, and that prison was an awful-but-necessary place that usually made people more violent and immoral than before they were arrested. Robert Martinson's now infamous (Martinson 1974) metanalysis was instrumental in the implementation of what would come to be known as the doctrine of "Nothing Works," which stripped recovery, rehabilitation and educational programs out of prisons and classified all such projects as wasteful. Movie producers reflected the cultural change in films that painted the prison system as a trap requiring extraordinary efforts to navigate. In 1974, *The Longest Yard* won a Golden Globe Award for best picture in the category of musical or comedy by telling the story of redeemed outlaw Paul Crewe, a professional quarterback who is sent to prison for drunk driving and theft, and who earns his redemption by refusing to intentionally lose a football game against the prison guards even when he is threatened with trumped up charges and additional punishment (Aldrich 1974). In 1979, *Escape From Alcatraz* followed outlaw Frank Morris into Alcatraz Federal Penitentiary, where he is accosted by both archetypal lawmen and other incarcerated characters, eventually forcing him to escape (Siegel 1979). In 1981, it was John Carpenter's (1981) *Escape from New York* with outlaw Snake Plissken, who earns his redemption by saving the President from a dystopian death. These films all reflected the prevailing cultural attitude that prisons were destructive, if necessary, spaces where likeable characters often wind up due to forces beyond their control, and where escape is often the only means of avoiding violence. Despite these changes in tone, the archetypal outlaws and lawmen remained, occupying their archetypal prisons.

The failure of prison to rehabilitate remained a popular theme throughout the 1990s and into the 2000s. Perhaps the most infamous prison film of all time, *The Shawshank Redemption*, premiered in 1994, pitting outlaw Andy Dufresne against the prison's obstinate lawmen, including the warden and his gang of dirty guards (Darabont). *Dead Man Walking* premiered in 1995, introducing archetypal outlaw Matthew Poncelet, who earns his redemption by eventually confessing to a crime he has long denied, but only as a result of his impending death, not from any form of rehabilitation or personal growth (Robbins). In 1996, *Sling Blade* put a spin on the idea of redemption when released outlaw Karl Childers winds up back in jail for killing a local woman's abusive boyfriend to protect her and her son, effectively channeling his unresolved criminality toward an acceptable target while revealing the prison's ultimate failure to rehabilitate him during his first stay (Thornton 1996). Decades later, producers would utilize the same trope of the outlaw channeling their desire for murder toward acceptable targets to earn a sort of redemption with shows like Showtime's *Dexter* (Cerone et al. 2006) and HBO's *Barry* (Berg et al. 2018).

In 2000, *Oh Brother, Where Art Thou?* rebooted Homer's Greek classic *The Odyssey* into a 1937 Mississippi setting with a story that follows three redeemed outlaws who escape a chain gang and are nearly killed by the obstinate lawman even after he receives word that they were pardoned (Coen 2000). In 2001, *The Last Castle* told the story of General Eugene Irwin, who is court-martialed and sent to a maximum-security military prison run by an obstinate lawman who eventually murders him (Lurie 2001). In 2002, Omar Epps starred in *Conviction*, a film based on the life of outlaw Carl Upchurch, who earns his redemption by organizing a nationwide peace summit for US gangs even as the local police act as lawmen who continue to harass and arrest him (Sullivan). In 2005, FOX's television series *Prison Break* hit prime time with redeemed outlaw Michael Scofield, a character who earns his

redemption by breaking his innocent brother out of a corrupt prison (Scheuring). In 2006 it was *Let's go to Prison!*, a slapstick comedy that pits redeemed outlaw Nelson Biederman IV against a prison full of carefree archetypal lawmen and unrepentant villains who run the place (Odenkirk 2006). In 2009, Jim Carey and Ewan McGregor played redeemed outlaws in the comedy film *I Love you Phillip Morris*, yet another representation of prison as a space where abuse is common and unpunished, and where escape is the only real option for redemption (Requa).

By the 2010s, race, class and gender were shaping cultural attitudes and academic scholarship related to crime and punishment. Michelle Alexander's (2010) *The New Jim Crow* was released in 2010, ushering in an era of critical research aimed at understanding and undoing the cultural stigma of incarceration while acknowledging the ways one's identity informs one's experience of punishment. Jenji Kohan's (Kohan 2013–2019) blockbuster *Orange is the New Black* was released in 2013, introducing redeemed outlaw Piper and a long list of obstinate lawmen who treat her and everyone else in the prison so poorly that a riot breaks out in season four. Kohan explained the show's success by pointing to Piper's whiteness in an interview with NPR's Terry Gross:

> In a lot of ways Piper was my Trojan Horse. You're not going to go into a network and sell a show on really fascinating tales of black women, and Latina women, and old women and criminals. But if you take this white girl, this sort of fish out of water, and you follow her in, you can then expand your world and tell all of those other stories. (Gross 2013)

Throughout these various updates in cinematic crime stories, the archetypal outlaw and lawman have remained stoically unchanged. There is, in every story of crime and punishment, an outlaw who seeks some sort of redemption, and a lawman who will not offer it.

The archetypal outlaw and his counterpart, the lawman, have been repackaged and resold as different characters in "new" movies for more than a century. Despite their apparent idiosyncrasies, each portrayal continues to display similar characteristics. We know them so well that introductions are not necessary, a boon to writers who are spared the hassle of drawing those lines in for us. Contemporary television shows and films that focus on imprisonment have also fostered the evolution of a cliché prison setting. The success of these films usually depends on the ability of their creators to emphasize the macabre while minimizing the banal, thereby painting a skewed picture of the prison industrial complex. We do not tune in to prisons on screen to see normal people living boring lives; we turn in to see the monsters in our nightmares do awful things to one another in confined spaces far away from our living rooms.

## 3. The Archetypal Prison

The outlaw and the lawman are in place, but the stage must still be set. In prison cinema, that stage is normally designed to fit one of three archetypes, which I shall refer to as prison-as-penance, prison as a playground, and prison as a paradox. Like the characters we meet inside these cinematic spaces, the differences which appear to set each production apart as original often disappear under scrutiny. Additionally, like those who occupy them, the prisons we see on screen give us tools for understanding our world. Unfortunately, the restricted nature of actual prison spaces makes it difficult for viewers to compare what they see on screen to real life, leaving us reliant on representations to make sense of our world. Given Hollywood's propensity to feed us what we demand—violence, mayhem and drama in the case of prison cinema—it is only natural that we come to view prisons as horrific institutional spaces required in any functioning society, and as such, as the only thing standing between us and the monsters we see on TV (Yousman 2009).

### 3.1. Prison-as-Penance

> "I completed all the programs. I followed the rules. But I still wound up back in prison."

I heard this story dozens of times during my incarceration, and I continue to hear it from my incarcerated students today. Given our ongoing crisis with recidivism, it's easy to see why. The most recent data is in line with US history: 82% of all released prisoners were rearrested within 10 years of their release from prison, and 61% of all released prisoners wound up back in prison between 2008-2018 (Antenangeli and Durose 2021). In the United States, redemption has become the exception, not the rule. But those who find themselves stuck in the revolving door of criminal corrections seldom fit the mold of stereotypical monsters dreamed up by Hollywood. The Bureau of Justice Statistics has repeatedly found that nearly half of all incarcerated people never receive a write-up for a rule violation during their entire sentence (Stephen 1989; Steiner and Wooldredge 2008). Incarcerated people frequently ask me to write letters of support to include with their requests for entry into overburdened educational and recovery programs. Prisons are packed with people who have been trying to earn redemption for decades only to be ultimately denied it regardless of what they do.

The trope of prison-as-penance portrays the penitentiary as a terrible-yet-necessary place where those who do awful things are sent for punishment. If they follow the rules and change their ways, outlaws in penance stories are eventually allowed to leave their past behind, while those who remain imprisoned indefinitely are products of their own poor decisions—bad guys of the worst sort designed to strike terror into the hearts of viewers. The ultimate punishment—death—is the natural endpoint of prison-as-penance narratives. Whether at the hands of state employees or others in prison, death looms over the heads of the incarcerated as a reminder that more punishment is always available if one continues to reject traditional redemption. The lawmen in penance narratives are depicted as good guys who protect the public from those who would otherwise run amok and endanger us all.

Redemption in a prison-as-penance story is earned through successful navigation of the prison space, which usually develops alongside a sincere sense of regret for one's crimes, an obvious change in one's ways, and, eventually, a devotion to helping others avoid a similar path in the future. Despite the violent prison environment with traps around every corner, the redeemed outlaw in a prison-as-penance story learns something from their incarceration that leads to a better life after release. Prison-as-penance reflects our cultural understanding of outlaws, lawmen and prisons so well that it shows up at some point in nearly all outlaw stories. Michel Foucault explained why in his groundbreaking work *Discipline and Punish*, where he traces the evolution of laws that were originally designed to exclude rule breakers from the larger social body through shame or death, but are now designed to bring rule-breakers back into the social fold by enticing (disciplining) them into conforming to social and cultural institutions (Foucault 1975). Prison-as-penance exemplifies Foucault's disciplinary apparatus aimed at producing self-regulation through technologies of power: "the expiation that once rained down upon the body must be replaced by a punishment that acts in depth upon the heart, the thoughts, the will, the inclinations" (p. 16). The result is a change of "heart" leading to a new performance of identity more in line with socially acceptable standards: prison-as-penance.

Prison-as-penance is more than just boring cells and squeaky bars. Often what the outlaw experiences in prison cinema goes far beyond what we would normally consider disciplinary, from torture to sexual assault sustained at the hands of other incarcerated people, to outright murder. When the end result of such stories is a change of heart, viewers are shown an image of prison as a successful tool of correction; it might be rough (at times even torturous), but at least it is effective in forcing some bad people to reevaluate their morals. Additionally, in those rare cases when even the worst punishment available fails to change the outlaw's ways and murder is not an appropriate option, prison-as-penance is always willing to cage the worst bad guys until the day they die, protecting viewers from the characters on screen who might otherwise haunt us in our sleep.

In *Conviction*, outlaw Carl Upchurch completes his prison sentence and walks out a changed man, abandoning his prior gang associations and eventually hosting a nationwide peace summit in Kansas City (Sullivan 2002). In *Dead Man Walking*, prison-as-penance

plays out as the main character nears his execution date and finally confesses in the 11th hour, painting prison and its ultimate end game—the death penalty—as effective tools of closure when nothing else works (Robbins 1995). In *Short Eyes*, penance is delivered not by state officials, but by those in jail when a man who is guilty of abusing children is about to be released because of faulty evidence, causing the outlaws-turned-lawmen to step in and enforce punishment (murder) where the system is incapable (Young 1977). In *American History X*, violent sexual assault at the hands of other incarcerated people initiates an unexpected change of heart in the main character, who then proceeds to fight against Neo-Nazi culture once released (Kaye 1998). In *The Green Mile*, we meet "Wild Bill" Wharton, a man who celebrates his upcoming execution and even swings from the bars screaming, "he's frying now!" while his neighbor is in the electric chair (Darabont 1999). Wharton reminds us that even though bad guys beyond redemption might exist, the prison system always has an answer, up to and including death. When we are shown image after image of prison as a place that keeps us safe from creepy characters who might otherwise do us harm while rehabilitating those who are willing to change, what else can we do but come to the conclusion that prison is a vital component for a functioning society? Without that preexisting belief, such narratives would not make sense (Yousman 2009). Without the constant flow of "new" narratives, such beliefs would not be sustainable.

### 3.2. Prison as a Playground

Prison is many things, but most of them are boring.

The first sixty days were the worst. Classified as "maximum security inmates" until the prison administration completed a personalized security-threat evaluation, we were locked up for 23 h each day. We left our cells for chow, for medical callouts, and for nothing else. Books were hard to come by and seldom included both halves. Mail and phone calls were restricted until we were classified and housed at a permanent facility. Property was scarce because most of us were new arrivals with nothing but the clothes and bedding handed out by the state when we arrived. Jackson was the absence of stimulation taken to the extreme. Any movement outside what was requested was forbidden, and anyone caught breaking the rules was immediately reassessed to a higher security level. There are no playgrounds in prison.

In representations of prison as a playground, the rules that restrict the movements and possessions of those in prison do not apply to a subset of incarcerated characters, often the worst behaved in the facility. Playground depictions showcase incarcerated shot callers who are not only immune to the regulations, but who can navigate the space of prison and use it to achieve their often-wicked ends. Leonidas Cheliotis described the fictional prison setting as a place where:

> Imagination tends to be taken on a sensational journey into spaces where the false and the fictional arise victorious from the ashes of the real. Prisons are usually typecast either as dark institutions of perpetual horror and virulent vandalism or idyllic holiday camps offering in-cell television and gourmet cuisine on the back of taxpayers. (Cheliotis 2010, p. 175)

Playground depictions are on full display when sexual assault appears commonplace and unpunished, when restricted items are easy to get, when officers can be bribed or blackmailed with ease, when escapes are carried out without much trouble, or when orchestrated hits are put out against those in prison. Many of the characters in prison as a playground thrive in the prison space, unbothered by the minor restrictions that do exist, and always ready to commit additional crimes should an opportunity arise (it always does). As viewers, we come to accept these depictions as representative of prison (Yousman 2013; Brown 2009). We might believe them to be exaggerated or glamorized, but we also view them as based on truths (De Certeau 1984; Yousman 2009). When the monsters on our screens live in taxpayer-funded facilities where they can continue committing crimes and victimizing others, how else can we respond except with disgust at the idea that prisons simply are not tough enough? When guards regularly ignore their own safety to smuggle

in cellphones, drugs or even weapons, what else can we do but desire harsher security and more restrictive rules (Yousman 2013).

Representations of prison as a playground are the bread and butter of movie producers. In Shawshank, Red is the guy "who can get you anything" (Darabont 1994). In *Goodfellas*, Paulie has a stockpile of steak, lobster, cigars and liquor, all served alongside "meatballs made with three kinds of meat: veal, beef and pork" (Scorsese 1990). As he explains, "wise guys" do not really mind being sent to prison: "everyone else in the joint was doing real time, all mixed together living like pigs . . . we owned the joint." *I Love you Phillip Morris* is packed with playground scenes, including men masturbating in public spaces and having sex without restrictions, contraband being bought, sold, and handed out as gifts, and even escapes that are so easy to pull off they become the central theme of the film (Requa and Ficarra 2009). In *Prison Break*, the main characters, who eventually break out, manage to bribe guards, possess restricted items (including weapons), move around the prison freely, dig a hole in the floor of the officer break room, and constantly assault one another at liberty (Scheuring 2005). In *Breaking Bad*, meth-kingpin "Heisenberg" manages to have ten people who are locked up inside three different prisons killed within the span of just a few minutes (MacLaren 2012). Prison as a playground paints an image of the criminal justice system wherein incarcerated people are seldom contained (or containable), where correctional officers have neither the desire nor the ability to prevent them from committing additional crimes, and where anything goes once one is in prison, a characterization that is dialectically opposed to the way most folks feel that prisons ought to be managed.

Much of the violence in representations of prison as a playground is sexual in nature, with the victimizers perpetrating their crimes at will and without penalty. There is a long cinematic history of anti-LGBTQ+ sentiment; as hooks explains, "Homophobic construction of gay sexual practice in mass media consistently reinforce the stereotypical notion that gay folks are predators, eager to feast upon the innocent" (Hooks 1994, p. 16). It is even worse in prison cinema, where viewers often expect to encounter a special kind of evil. It is so common to see sexual assault in depictions of correctional environments that we are seldom surprised when such storylines play out. Helen Eigenberg and Agnes Baro found that "while most studies on male rape in prison suggest that it is a relatively rare event," in contemporary movies, "the inclusion of at least a reference to male rape and/or a peripheral rape scene has become a standard part of prison film production" (Eigenberg and Baro 2003, pp. 64, 86). Dawn Cecil notes a similar thread running through women's prison films: "These 'babes-behind-bars' films perpetuate highly sexualized images of female prisoners . . . they do not necessarily seek to educate—instead they aim to titillate" (Cecil 2007, p. 305). In *Prison Break*, a character known as T-Bag regularly dominates others by turning his pockets inside out and forcing victims to hold them while following him around the yard, a move that would get you written up in any real prison (Scheuring 2005; Federal Bureau of Prisons 2012). In 2002, soft drink giant 7-Up ran a commercial where a man hands out cans of soda between cell bars until he drops one and quips, "I'm not picking that up," a shout-out to the trope of sexual victimization around every corner (Walker 2002). In *Barbershop*, when J.D. is told that someone else is going to prison for his crime, he quips, "don't drop the soap" (Story 2002). In prison films, sex sells, especially if its uninvited. In actual prisons, the rate of sexual assault hovers around 1.3% of incarcerated people per year (Buehler 2021). The spectacle is misleading, to say the least.

The extreme storylines offered in prison cinema would not work if the space was not already stocked with the bad guys Hollywood dreams up to turn out stomachs. Sexual assault is commonplace and unpunished in playground films, but it only happens because many of those inside enjoy the sexual victimization of others. Prison riots and violent murders only work when there are plenty of characters to do the assaulting. And those characters are often superhuman in their strength, endurance and cruelty. In *Fast & Furious 8*, ex-cop Luke Hobbs gives us an image of prison as a playground when he is locked in the same maximum-security facility as his archrival, Shaw (Gray 2017). The action in this playground for hyperviolent men reaches a crescendo when Hobbs and Shaw are both

unexpectedly released from their cells and, of course, immediately begin fighting with each other, along with dozens of taser-wielding guards dressed in body armor. Since the trope of prison as a playground relies on evil monsters who are always a threat to social order and nearly impossible to subdue, producers opened all the cell doors along with those of the two main characters, setting their cinematic monsters free. Unsurprisingly, all-out war ensues when everyone comes charging out to immediately attack officers and one another, because that is what incarcerated people are like in the movies. A stereotypical Hollywood prisoner is not only hyperviolent, but also hard to contain and incredibly strong, a trope performed by Hobbs when a guard shoots him with a beanbag shotgun and the bullets bounce off his chest without injury. He quips, "beanbag shotgun; big mistake," then throws one guard across the room and shoots another with his own gun.

Prison as a playground tempts viewers to stay tuned in with stories about our worst fears come to life. The redeemed outlaw in a playground film is stuck in the box of penance, where they must navigate the penitentiary and survive constant threats to safety and integrity. However, one man's penance is frequently another man's playground, and producers have learned well that viewers eat up the drama of unwritten rules, confusing victimization, and counterproductive rehabilitation in prison films. Prisons on screen do not contain or reform their residents; they make them more dangerous, antisocial and heartless. Enter the trope of prison as a paradox.

### 3.3. Prison as a Paradox

"You are Inmate Boyce, 470236? Is that correct?"

Parole hearings are the most terrifying part of incarceration for most people, far more threatening than showers, the chow hall, or the prison yard. The three-person panel was tasked with deciding whether I deserved a shot at early release—at redemption—and they were well prepared. Each member had two folders, a thick one packed with my criminal past of arrests for hustling and stealing, and another one, thinner than the first, that contained my institutional history. I had just a single ticket over the course of the prior year, and as one member quickly pointed out, I had managed to complete one of the prison's only educational programs—a typing class. I'd also forwarded them a letter confessing my addiction and expressing my sincere desire to continue my recovery after release. I was only 25 years old, and I'd been locked up for a few of those years, so despite an impressive record of convictions, they'd all come over a relatively short period of time. The board said a few more words about their concern for community safety and "offender rehabilitation," then they sent me on my way to make their decision in private.

A few weeks later, my release papers showed up; I'd been deemed an acceptable risk.

According to the Federal Bureau of Prisons, incarceration should serve the two-fold purpose of threat containment and threat reduction (rehabilitation):

> ... to protect society by confining offenders in the controlled environments of prisons and ... [to] provide work and other self-improvement opportunities to assist offenders in becoming law-abiding citizens. (US Federal Bureau of Prisons 2022)

The trope of prison as a paradox flips these stated goals to their opposites, painting prison as a space that *expands* threats and *increases* criminality. The end result of paradox narratives is an understanding of prison as a dysfunctional institution that worsens any problems it claims to solve.

Prison cinema often plays on all our worst fears and our deepest frustrations with human nature. On-screen prisons are not just cages. Once the script is memorized and the cameras are rolling, prisons become magical spaces of terror and spectacle where viewers can travel at will without fear of physical harm or emotional trauma.[2] In those spaces, we are primed to accept the bizarre and expect the extreme. Aiming to please, bizarre and extreme is exactly what producers and writers feed us, knowing we are unlikely to challenge storylines that would strike us as unbelievable if set anywhere outside of prison. As a result, we often come to understand prison as a paradox: as a place where people who

commit crimes are sent for rehabilitation only to wind up more prone to criminality and victimization than before they were arrested.

In *The Shawshank Redemption*, the warden recruits the outlaw as an illegal accountant, showing an image of prison where crime is not only allowed, but encouraged by those in charge (Darabont 1994). Both *Life* (Demme 1999) and *Cool Hand Luke* (Rosenberg 1967) include fight scenes where the main outlaw is beaten by one of the largest men in the prison in full view of prison officials, providing an image of prison as a place where violence is encouraged and fighting is normalized. In *Short Eyes*, the prison administration fails to prevent incarcerated characters from committing a group murder, showing an image of jail where an arrest for minor crimes can inevitably lead to participation in organized killing (Young 1977). By season three of *Oz*, Tobias Beecher, who is sent to prison for a drunk driving accident resulting in someone's death, has been transformed from a peaceful man into a throat-slashing murderer, painting prison as a space that, as Damien Echols has said, can "send a man to prison for writing bad checks and then torment him there until he becomes a violent offender" (Fontana 2003; Echols 2012, p. 14). Within five minutes of arriving at his first prison in *Blood In Blood Out*, young outlaw Milko is sexually assaulted and forced to join a gang that eventually crafts him into a stereotypical violent recidivist (Hackford 1993). People do not go to prisons like these to become contributing members of society; they go there to learn new tricks and experience new traumas. The paradoxical nature of such scenes leaves viewers with the impression that those who have been to prison are worse when they get out than they were when they went in.

Prison as a paradox reminds us, wrongly, that despite its claims of containing threats and rehabilitating redeemable outlaws, prison often increases the social threat of criminality. Representations of prison as a twisted game where villains enforce confusing rules and victimize others for unclear reasons paint the correctional space as fraught with danger around every corner and incapable of offering effective rehabilitation of any sort. For the victimizers, prison is painted as a playground for sadistic behavior. For their victims, prison is painted as torturous penance. However, for the viewer of such stories, prison is transformed into a paradoxical trap—a place where the incarcerated must learn to victimize others if they are to avoid becoming victims themselves. Any time characters inside a cinematic prison flaunt the rules to constantly hurt one another, viewers must contend with the paradox of an expensive prison system that neither contains threats to social order nor rehabilitates offenders, the two claimed goals of criminal corrections. Unlike representations of prison-as-penance or prison as a playground, prison as a paradox opens a space for viewers to reconsider the effectiveness of the criminal justice system. That is why producers usually slide out of paradox representations and back into the more comfortable ground of playground or penance by the conclusion of most stories.

## 4. Updating the Spectacle

We learn from media whether we want to or not. What we learn comes to inform our perspective of the world, and our ability to understand it and explain it to others (Yousman 2009, 2013; Hooks 1994; Dixon 2017; Donovan and Klahm 2015). When the spaces we learn about on screen are restricted from public oversight, the spectacle becomes conflated with our understanding of the world. Even in spaces where most of us can easily verify what we see on television or film, we usually do not bother to do so. How often have you looked for the hatch in the roof of an elevator only to realize it does not exist, glanced at an air vent in an office and thoughts, "a human couldn't crawl through that," or witnessed an actual police standoff and realized they do not yell, "cover me!" and run into the line of fire? Probably more than once, and those are spaces many of us can access on a regular basis. All the worse when it comes to prisons, which most of us never have the opportunity or the desire to enter in person.

Our cultural understanding of prison and incarceration is aligned with our cultural demand for updated spectacles and repackaged stories of mayhem and murder. We learn about drugs, crime and punishment from our televisions and movies, and what we learn is

often a far cry from reality. Our media reflects a culture obsessed with mass incarceration, which has become a permanent staple of US life. Any change in our cultural understanding of crime and punishment must begin and end with a cultural update in our representations of crime and punishment. Until we come to see those in prison as worthy of redemption and capable of achievement, we will continue to demand the same stereotypical spectacles. And until the spectacles are updated to reflect our demand for humanity, we will continue to view those behind bars as occasional outlaws amongst hordes of villains.

What, then, can producers do to draw an audience? If mayhem and murder are what sells, how can we expect salespeople to peddle less popular products? The answer is not to do away with the spectacle, but to update it. The revolution has already begun, and it does not require irredeemable ghouls to keep us tuned in.

The world of podcasting offers some rich examples. The award-winning *Ear Hustle*, hosted by people who are (or who have been) incarcerated and recorded inside San Quentin Prison, currently has nine seasons available for download. *Ear Hustle* is devoted to sharing the stories of incarcerated people who are working for change. Episodes feature true-crime narratives from guests, along with the real stories of those in prison: their histories, their accomplishments, and their ongoing attempts are redemption. The University of Denver Prison Arts Initiative has been recording and releasing episodes of its prison-based podcast *With(in)* since 2019, focusing on the stories and pathways to success of those inside Colorado prisons. The topics might at times be titillating and salacious, but they are also personal and humanized, told by those who lived them and expanded beyond the criminal charges which the world often uses to define those of us who have been to prison. Such productions feed our desire for the taboo, but they do so in a way that paints those in prison as humans who made mistakes and who are capable of redemption. They are part of a new wave of productions that present what Nellis has called "The newly emergent 'convict criminology,' produced in the USA by ex-cons (usually imprisoned as a result of the 'war on drugs') who have subsequently become academics" (Nellis 2009, p. 144).

New prison narratives like *Ear Hustle* and *With(in)* also focus on the various ways the current design of the system makes redemption difficult while sharing stories of those who have overcome the odds against them and worked to change their lives. Of course, many of us manage to avoid the trap of recidivism after we are released. Many of us, myself included, do more than that. We build careers. We earn degrees. We reenter prison as educators, counselors and program directors. And all of it has an air of drama—that Hollywood magic that comes with prison walls and razor wire on screen. We can have it both ways; we can demand productions that humanize those in prison while also enjoying the interesting stories about how they got there, and how they get out.

**Funding:** This research received no external funding.

**Institutional Review Board Statement:** Not applicable.

**Informed Consent Statement:** Not applicable.

**Data Availability Statement:** No new data were created or analyzed in this study. Data sharing is not applicable to this article.

**Conflicts of Interest:** The author declares no conflict of interests.

## Notes

[1]  The film versions vary from one another. For example, the 1978 version depicted Valjean escaping from Toulon Prison, not gaining his parole, and he spends his life fleeing from the original charges, whereas the 2012 version depicts Valjean breaking his parole and creating a new identity in response to the gift of silver. The original novel begins with his parole, not his escape.

[2]  Understandably, many of us cannot travel to the space of prison film without (re)experiencing trauma.

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
