# Peer review of "Lessons from Shawshank: Outlaws, Lawmen and the Spectacle of Punishment"

_humanities, doi:10.3390/h12010010_

Round 1
Reviewer 1 Report
The paper does not begin with a clear position on the matter of the need for updated portrayals of incarceration. The paper would benefit from a clear thesis at the beginning. This thesis would help to give the paper a more linear and logical structure to the argument.
Author Response
Reviewer 1 Comments
Reply: I added a paragraph after the autobiographical introduction that previews the goal of the paper and offers a thesis—a call to reorient the cinematic space of prison film production toward updated spectacles of punishment more conducive to the goal of destigmatizing the stigma of incarceration.

Reviewer 2 Report
The piece sets out to explore the question of why Americans know so little about prisons and the criminal justice system even though films and television shows seem to explore those systems so frequently. The question is an important and pressing one amid the calls for "defunding the police" and "prison abolition."
I did wonder if this piece feels more like a book project than a single article. It is trying to cover a lot of territory in a fairly short article. The result is that this piece sometimes feels more like an overview than the kind of detailed examination I might expect in a scholarly article.
The first section seems like it moves very quickly through the Frankfurt School, recent scholarship about crime dramas, and the economic reasons why television and film increasingly relies on established intellectual properties for the basis of shows. All of this is heading in the right direction, but it seems like there is more to be said in these sections than a paragraph allows. There also is a bit of a conflation of crime dramas and texts focused on prisons. To this reader, they feel like separate genres. More on this below.
The rest of the piece seems like it wants to go in two slightly different and somewhat opposing directions. The first part of the article sets up the contrast between two archetypes: the outlaw and the lawman. This section is well-done and insightful. The use of Les Miserable seems to work well for this section (although I am an americanist and only have a general knowledge of this text). It seems like a longer version of this project would have done more with this contrast, explaining its cultural context, meaning, and persistence over time and in various cultures. This section also seem to have as much or more to say about crime dramas as it did about prison literature and films, per se.
The essay then tries to build a history of these archetypes that seems to jump around time and place. I like where this is going and found this very intriguing, but it blurs eras, cultures, and shifts in cultural attitudes in time. While I do think there is an interesting "tradition" here, it feels like this recitation does not do justice to the context of these specific pieces. For example, The Bird Man of Alcatraz seems to have more in common with Gideon's Trumpet (not discussed here) and the civil rights revolution of the 1950s and 1960s, where there was more optimism about courts and the legal systems than texts like The Shawshank Redemption and Dead Man Walking that come from a much more critical place of the 1990s, rooted in critical legal studies and critical race theory. These nuances would deepen the analysis and strengthen the author's point.
The last part of the paper makes (once again) some fascinating observations about various prison archetypes. I found the sections about prison as penance and prison as playground clear, coherent , and fairly well-supported (although they both could benefit from historicizing and contextualizing the analysis.) I though the last section, "prison as paradox," was not as well developed or coherent. The "paradoxes" did not seem to meet the definition of the word. Yes, the examples are confusing, incoherent, or simply inaccurate representations of the experience of prison, but the article needs to make the actual paradox or paradoxes clearer. For example, the lengthy and persuasive paragraph (463-486) about the use of the fish metaphor shows its ubiquity. What it doesn't show is a paradox.
Lastly, I didn't think the conclusion (488-510) added much to the essay. It doesn't really explain the implications of the work, other than a sense that the work of prison reform must include a revision of prison images in popular culture. I thought that was a weak ending because the images of incarcerated persons are the result of political, cultural, and racial forces. This piece doesn't do enough to explain those forces. For example, would Shawshank Redemption been made if Morgan Freeman played Andy and Tim Robbins played Red? Or does the book/film come out of specific cultural moment in how it deploys these archetypes? I also thought the ending might return to the autobiographical introduction and offering a more "realistic" or mundane sense of prison life.
I do support a revised version of this article being published, but I do hope the author makes some effort clarify the scope and purpose of this essay and offer more of the cultural context for these representations. I will be honest though that I would actually be more excited for book-length manuscript that further develops these ideas than seeing only a revised essay getting published!
Author Response
Reviewer 2 Comments (in order of suggestions). Specific replies to each comment included in the attached file with original comments.
Reply: I attempted to flesh out the Frankfurt School by linking it to Hannah Arendt’s work regarding Eichmann’s trial in Jerusalem and her conclusion that he (like everyone) was influenced by mediated messages. I also added cultural markers throughout: 1960s Civil Rights Movement, 1970s "Nothing Works" mentality, 1990s tough on crime era, 2010s The New Jim Crow critical scholarship, etc.
For prison as a paradox, I have updated the entire section to emphasize the paradox of prison as a rehabilitative, punitive space where people go for rehabilitation and containment, but wind up uncontained and more criminal than before arrest. I have also cut some of the heavy doses of examples from multiple movies in favor of individual examples which I take the time to more fully connect to the paradox.
The conclusion now includes some brief references to updated spectacles that might serve as frameworks for future attempts to preserve cinematic success based on old recipes of excitement and true crime while avoiding the traps of stereotype reinforcement. Specially, the space of podcasting is where I point, using Ear Hustle and With(In) as two examples of what might be called updated spectacles of punishment.

Reviewer 3 Report
This is a clearly written and interesting essay with a lot of potential to be a fascinating, highly innovative, and influential article if you can make your own prison experience more central to your argument, using your expertise on the subject both to a) assess the accuracy of movie prison (mis)representations and b) build on other scholars’ representational theories. The high point of the essay for me is at the bottom of page 6, where you write about the contrast between what viewers expect in screen representations of prison and the reality of “normal people living boring lives” behind bars. A paragraph at some point on the banality of prison life would be welcome—a description of the kinds of routines that characterize the average, humdrum penitentiary days that non-incarcerated people rarely or never hear about. But at many points throughout the essay, I think your argument would benefit from more explicitly stated reasons and additional evidence regarding why particular prison archetypes are wrong, or only partly true. I certainly understand if you’re eager to avoid making this an overly autobiographical essay, but it wouldn’t be hard to frame your remarks in a relatively objective way that nonetheless conveys the idea that you’ve “done your homework” as very few other scholars have. For instance, when you’re talking about fictional prisoners who flagrantly break the rules and have unlimited access to weapons, drugs, and so on, you could say something about why it would make no sense for guards and wardens to put their own jobs—and lives—in jeopardy by allowing this kind of thing to happen. (That comment is based on conversations I had with a guy who worked as a prison guard in California, but forgive me if I’m overgeneralizing based on what he told me.) While incarcerated, did you talk with other inmates about TV and movie representations of prison life? Were you able to watch any of the shows and films you’ve covered while you were actually in prison? I’ve watched no shortage of these fictional representations myself, but what I’ve never encountered is a point-by-point critique of them by the true experts, the people who are the real-world prototypes of the fictional characters, not just asserting that the representations are warped but showing what they get wrong by contrasting fiction with reality. Additional references to empirical studies on sexual violence, the treatment of new inmates, prison escape rates, and so forth would be helpful in this regard. (By the way, I really like the parts of the essay in which you explain what non-incarcerated viewers get out of these fictions.)
To elaborate on point b) above, I think evidence drawn from your personal experiences—however objectively framed—could likewise do a lot to enrich theories of prison representations. At the very least, I would like to see you highlight how your essay is challenging and building on the work of such scholars as Michelle Brown and Bill Yousman; more direct engagement with them from early on in the piece would do a lot to strengthen your critical framework. Which aspects of prison life do they overlook and/or over- or underemphasize? How did you see their points reflected/refracted in your daily life in prison and in the behavior of other people there? It would really help your essay stand out if you were to test their claims against the lived realities of incarceration. And if you could add (here and there throughout the essay) to what you say in your introduction about how misrepresentations of prison shaped your own attitudes and behavior before you were taken to the penitentiary, thereby articulating your own theory of prison representation from inside the carceral system, I think your essay could have a big impact on future writing in this field.
In terms of your engagement with theories of media effects in general, the essay could use both more nuance in its claims and more empirical support from other scholars. For example, when you write on page 3 that “When faulty information is the only information available, we come to accept fiction as fact,” I would like to see you bring some more critical pressure to bear on that “we” as well as deal with a counterargument or two and substantiate the claim with evidence from one or more studies showing how susceptible people are to misinformation, particularly in the form of fictional TV shows and movies.
Let me stress, again, that this essay has a great deal of potential. I wouldn’t have written so much if it didn’t. I don’t want to burden you with too many other suggestions for improvement, but here are a few comments about local issues in the piece:
Page 2: I like your emphasis throughout the essay on how producers keep recycling rather absurd, outworn tropes and archetypes and passing them off as bold innovations. I think it would clarify the sentence beginning in line 86 to say “Television is full of ‘new’ productions based on shows that were canceled decades ago…”
Page 3: To be honest, the paragraph on archetypes at the bottom of the page doesn’t seem to contribute much to your argument. I would recommend either cutting it or making it more debatable by talking about what other scholars haven’t noticed about these archetypes or about how they’ve mischaracterized the archetypes’ influences on viewers. (Something along those lines.)
Page 4: Using Valjean and Javert as prototypes is a great idea.
Page 5: Please explain the difference between outlaws who seek/find redemption in prison and innocent people who are sentenced to prison unjustly. Exoneration is undoubtedly a form of redemption, but trying to get out of prison when you’re innocent is, of course, very different from seeking forgiveness for a crime you acknowledge you’ve committed.
Page 7: I don’t know if you want to avoid bringing in Foucault’s Discipline and Punish here, but it would make a lot of sense when you’re talking about prisons-as-penance (and I recommend using the hyphens throughout your discussion) to add at least a few words on the history of penitentiaries and the etymology of the term. Perhaps you could argue that the prison-as-penance film archetype is influential, in part, because it grows out of much older versions of the archetype.
The end: I was expecting either a return to your own life story or a discussion of a film or show such as Rectify (maybe?) that tries, even if it fails, to do more justice to the complexities of the experiences of incarcerated people than the average representation. (Or maybe some combination of these two approaches would work.)
Anyway, I hope these comments are clear, and I look forward to seeing your essay get published!
Author Response
Reviewer 3: Full comments with original suggestions attached
Reply: I have added additional autobiographical entries at various turning points throughout the piece in an effort to anchor it to my own personal experience in prison, and to emphasize the schism between the real and the represented.
These were great suggestions, and I wound up weaving much of them throughout the essay, including auto-biographical accounts and additional research to bolster claims. Each section now begins with a brief autobiographical account of a prison experience that clashes with the spectacles discussed in that section. I also wound up including your suggestion about guard safety along with other obvious conflicts that viewers would probably notice right away if not for the cinematic rules of prison spaces, which cause us to suspend some of our better judgements.
I have added a few additional citations to this first section to provide some prison-specific examples from research about how representations teach us things without us realizing it.
In reference to the difference between incarcerated characters who the audience knows to be innocent and those we discover are innocent later in the story, I focus on just the latter for the purpose of this essay (largely in the goal of keeping it short enough to publish). I focus on the use of innocence as a method of rushing an audience to offer redemption. I added a sentence early into this section describing the use of this plot device.
I also provide some connections to Foucault's work in Prison-as-Penance.

Round 2
Reviewer 3 Report
These revisions are excellent, and I’m impressed that you were able to add the new parts so quickly. The new autobiographical passages represent a great contribution to your argument, making it more original and compelling, but I also appreciate the improved transitions and passages in which you spell the implications of your argument out more directly. I think the essay in this form has a better chance of influencing scholars’ thinking about prisons and prison representations than the first version. Thank you for taking my suggestions so seriously!
There are a few minor errors that can still be cleaned up, but I wanted to comment on more significant things—both strengths of this version and additional areas for improvement.
Page 3: The details in the new section are terrific. Having just read an opinion piece in The New York Times by Reginald Dwayne Betts about his Freedom Reads project, I especially like the fact that you bring up books. I can recall plenty of TV and movie representations of incarcerated people killing each other (and guards), scheming, rioting, smuggling in contraband, and so on, but I can’t think of a single camera shot of a prisoner reading a book—except, possibly, for Hannibal Lecter in The Silence of the Lambs. If our society is truly interested in helping incarcerated people achieve redemption, this seems like an egregious oversight, given that Betts and so many other formerly incarcerated people have credited books they read in prison with giving them a vision of a different life.
Page 4: There’s some confusion in the first paragraph about which version of Les Misérables you’re discussing. I would just say, “Since 1985, the musical upon which the film is based…”
Page 5: I don’t remember commenting on this before: your observation on Valjean and Javert’s role reversal is smart.
Page 6: I checked, and the title of Jailhouse Rock is formatted that way rather than as Jail House Rock. Also, I think you should mention Elvis, not just because he was so famous but because he was an eminently “redeemable” actor to cast as an inmate. The material you added on that page about changes in actual prisons and representations of them in the 1970s is really helpful.
Page 7: The transition hinging on Michelle Alexander’s book is great.
Page 8: The new section is powerful; I would just recommend specifying the range of years covered by Antenangeli and Durose’s study.
Page 10: You have added another vivid autobiographical section on this page. I might just add a brief explanation after the sentence reading “The first sixty days were the worst,” such as “Classified as ‘maximum security’ inmates because the prison administration wanted to…”
Page 12: I like the placement of this autobiographical passage, and the line about how parole hearings are the most terrifying thing about prison for most inmates contributes a lot to your argument. I can’t decide whether it would improve or diminish that sentence to add a few words to it in order to create a sharper contrast. Example: “Parole hearings—not dealing with guards, or resolving conflicts with other inmates, or…” Maybe it’s perfect the way it is.
Page 13: I think with your added evidence and your synthesis at the end of the section, you’ve done a good job of responding to the other reviewer’s comments about your handling of prison-as-paradox, but of course that’s for the other reviewer to say.
Page 14: The new ending is strong.
Author Response
Thanks again for the advice. I have added the line, "Since 1985, the musical upon which the film is based" to specify which version of Les Misérables I'm discussing. Jailhouse Rock is fixed and Elvis is included in that description. Antenangeli and Durose’s study dates (2008-2018) are now included. I've also specified both the reason maximum security is the norm for all new arrivals to prison, and the contrast of the parole board being more terrifying than the showers, yard and chow hall.